# Interactions with Arsenic: Mechanisms of Toxicity and Cellular Resistance in Eukaryotic Microorganisms

**DOI:** 10.3390/ijerph182212226

**Published:** 2021-11-21

**Authors:** Patricia De Francisco, Ana Martín-González, Daniel Rodriguez-Martín, Silvia Díaz

**Affiliations:** 1Astrobiology Center (INTA-CSIC), Carretera de Ajalvir Km 4, 28850 Madrid, Spain; pdefrancisco@cab.intacsic.es; 2Department of Genetics, Physiology and Microbiology, Faculty of Biology, C/José Antonio Novais, 12, Universidad Complutense de Madrid (UCM), 28040 Madrid, Spain; anamarti@bio.ucm.es; 3Animal Health Research Centre (CISA), National Institute for Agricultural and Food Research and Technology (INIA-CSIC), 28130 Madrid, Spain; rodriguez.daniel@inia.es

**Keywords:** arsenic, toxicity, yeasts, microalgae, resistance mechanisms

## Abstract

Arsenic (As) is quite an abundant metalloid, with ancient origin and ubiquitous distribution, which represents a severe environmental risk and a global problem for public health. Microbial exposure to As compounds in the environment has happened since the beginning of time. Selective pressure has induced the evolution of various genetic systems conferring useful capacities in many microorganisms to detoxify and even use arsenic, as an energy source. This review summarizes the microbial impact of the As biogeochemical cycle. Moreover, the poorly known adverse effects of this element on eukaryotic microbes, as well as the As uptake and detoxification mechanisms developed by yeast and protists, are discussed. Finally, an outlook of As microbial remediation makes evident the knowledge gaps and the necessity of new approaches to mitigate this environmental challenge.

## 1. By Way of Introduction

This review has been structured in seven different parts, ranging from general concepts of arsenic (As) contamination to more specific knowledge of microorganism–As interactions. Different characteristics of the impact, importance, and complexity of As, as a priority pollutant, which has compromised the health of more than 200 million human lives, are highlighted. The toxicity of the different chemical forms of As on microorganisms, as well as their mechanisms of resistance to this metalloid, should serve to under-stand the role of microbes in the biogenic cycles that are intimately involved in the recycling of As and its application in bioremediation. In fact, we analyzed and updated some relevant aspects of arsenic contamination on our planet, such as its environmental impact and health risks. This crucial environmental problem has complex solutions, due to the complex chemistry and biochemistry of this metalloid, as we have explained in part 2 of the review. Global As contamination is a consequence of geogenic sources and mainly of multiple anthropogenic sources (industry, mining, chemotherapy, etc.). The sources of As and their impacts were considered in parts 3 and 4 of this review. As is an element that is continuously recycled in the environment. Abiotic factors (environmental and geological) modulate the physiology and distribution of microorganisms, mainly prokaryotes (Bacteria and Archaea) that are involved in changes among As species and, in addition, change some of these abiotic factors. Numerous studies show that microbial interactions with As have an ancient origin. In this context, biotransformations and As resistance mechanisms are poorly understood in eukaryotic microorganisms (yeasts, microalgae, and ciliates), despite their importance in both As recycling and, especially, biological immobilization of As. All these mechanisms have been described and updated in parts 6 and 7 of the review. Finally, some personal reflections and recent data on new potential avenues for microbial As remediation are presented.

## 2. Arsenic, a Metalloid with Complex Chemistry and Biogeochemistry

Arsenic is a naturally occurring metalloid rather abundant in the earth´s crust (0.00015%). It is reported as the 12th most abundant element in the human body, the 20th on the earth’s surface, and the 14th in seawater [1,2]. Moreover, it is present in more than 300 minerals, of which 60% are in arsenate form, 20% are sulfides and sulfosalts, most of them combined with other elements (Cu, Fe, Ag, etc.), such as orpiment (arsenic sulfide), enargite (copper arsenic sulfide), or arsenopyrite (iron arsenide sulfide), and the remaining 20% are in the form of arsenites, arsenides, silicates, oxides, and elemental As [3,4]. As chemistry and biogeochemistry is complex and can be influenced by physico-chemical, geological, and biological factors. As can exist in four different oxidation states: As(-III), As (0), As(III), or As(V). Figure 1 shows the main chemical forms of arsenic. The trivalent arsenic [As(III)] and the pentavalent arsenic [As(V)] are widely present in natural waters, due to their solubility over a wide range of pH and Eh conditions [5]. Under oxidizing, near-neutral conditions, such as those found in many surface waters, arsenic occurs predominately as As(V), whereas, under reducing slightly acidic conditions, such as those found in many reducing subsurface environments, it is often found as As(III) [6]. Arsine (AsH_3_) is a highly toxic inorganic arsenic species, but its reactivity to oxygen means that other species tend to be of greater importance in the environment [6,7]. Besides the most common inorganic forms [As(V) and As(III)], some biological systems can generate methylated arsenic compounds from As(III) and As(V), i.e., mononomethylarsonic acid (MMA), dimetrylarsinic acid (DMA), and trimethylarsine oxide (TMAO) [8]. In addition to As(III), As(V), and their methylated derivatives, a large number of organoarsenic compounds are found in the environment [9,10]. Some of them present important biological functions; like arsenobetaine, that confers bacterial and microalgal cytoprotection against osmotic stress and temperature extremes, and arsenocholine, that are found in diverse marine organisms, as some fishes and shellfish [11,12,13]. In general, it has been stated that inorganic As forms are much more toxic than organoarsenicals in microorganisms, animals, and plants [3,14,15]. Therefore, inorganic As is often biotransformed into organoarsenicals by distinct eukaryotic and prokaryotic microorganisms, presumably for detoxification or utilization, as an energy source [16]. All of these arsenic chemical forms are present in soils, superficial waters (freshwater and marine), groundwater, and even in biological systems (organisms and microorganisms) [5,17,18].

Concentrations and relative proportions of As species vary according to changes in As source, environmental conditions, and biological activity. Environmental conditions such as pH, temperature, organic matter content, humidity, and redox state, as well as biotic influences, will affect the toxicity of As by directly influencing bioavailability and speciation [19]. Redox potential and pH are the main environmental factors that control As speciation in waters [7,8,20]. The redox potential of arsenic oxyanions is very relevant, in such a way that As(III) becomes stable in aqueous form under moderately reducing conditions (+300 mV at pH 4 to −200 mV at pH 9), while As(V) is stable in oxidized aqueous solutions [21]. Additionally, As biotransformation by different physiological groups of microorganisms plays a significant role in the occurrence and behavior, as well as recycling of this metalloid in the aquatic environments [22,23]. In aquatic ecosystems, such as lakes, As(V) is the thermodynamically stable state in oxic conditions, while As(III) is predominant in reduced environments [24]. Under reducing conditions at a pH lower than 9.2, the neutral trivalent arsenic species H_3_AsO_3_ exists, which dissociates to form anions under high pH conditions only [18]. The occurrence of elemental As (As^0^) is rare in nature and most of it is produced by biological activity [17]. In soils, As species are more diverse, including inorganic, organic, and arsenic-containing minerals [1]. These different chemical forms of As can be found precipitated as solids, adsorbed by organic or inorganic soil constituents, free ionic forms, and finally as structural constituents of primary and secondary minerals [25].

## 3. Arsenic Sources and Emissions in the Biosphere and Atmosphere

As we will examine in detail later, microorganisms play relevant roles in the environmental fates of As since they can carry out different transformations, so in aquatic and environmental environments, there are continuous transformations between soluble and insoluble forms and therefore, between toxic and nontoxic forms [5,15,26]. Besides the microbial impact on ecosystems, As can enter in terrestrial and aquatic environments via both natural geogenic processes and anthropogenic activities (Figure 2) [18]. Known and potential natural sources of As include hydrothermal and geothermal emissions, hydrocarbon reservoirs, mineral ores, coal deposits, atmospheric dust and aerosols, dissolution of sulfide minerals, forest fire, and biological mobilization [27]. In the lithosphere, As is mainly associated with sulphide minerals. In the atmosphere, As dominantly occurs in the species arsine, metallic arsenic, inorganic trivalent and pentavalent arsenic, organic monomethylarsenic acid (MMA), dimethylarsenic acid (DMA) and/or their salts [28]. Of course, the anthropogenic sources have a qualitative and especially quantitative higher impact on the increase of As pollution in ecosystems. Moreover, anthropogenic activities play an important role in dispersing As contamination to the hydrosphere, pedosphere, and atmosphere [29]. As has been employed by humans for years in industrial practices, although most of them are not allowed at present by regulatory directives, due to their environmental and health risks. Thus, it has been used in the production of semiconductors, pigments, cosmetics, insecticides and herbicides, tanning industry, lead-acid batteries, in the glass industry, and copper refining industry, among others [30]. As is one of the priority pollutants associated with acid mine drainage, especially from gold mining operations. Moreover, the hydrometallurgical and pyrometallurgical processes applied for processing complex arsenic-bearing minerals has increased due to a decrease in the traditional base metal reserves [31].

In the first half of the nineteenth century, inorganic pesticides (herbicides, insecticides) were normally used in agriculture and were found to be stable in the environment having an affinity to water [32]. Chromated copper arsenate is a chemical additive used traditionally to preserve wood from decomposing, due to humidity, insects, and microbial agents (biodeterioration) [33,34]. Finally, arsenic has a long history as a human poison and paradoxically, as a therapeutic agent [18,35]. In ancient times, arsenic sulfides were used to treat ulcers and abscesses, and later (1200s), in the Middle East, for the treatment of skin diseases, hemorrhoids, and syphilis [36]. In the late 18th century, Fowler’s solution was discovered. This is a 1% solution of potassium arsenite that was used in the treatment of various diseases, including malaria, syphilis, asthma, chorea, eczema, and psoriasis. In 1910, Paul Ehrlich introduced a new arsenic-based drug to treat syphilis, the organoarsenic compound arsphenamine, sold commercially under the name of Salvarsan, which was used until the penicillin treatment became more prevalent in the 1940s [35,36,37]. More recently, some organic arsenical compounds, and particularly the inorganic form arsenic trioxide, are valued, well-researched, and effective chemotherapy agents for solid and disperse tumors [38,39].

Natural inputs of As to the atmosphere come mainly from volcanic activity, biovolatilisation, wind erosion of soils and salt dissolutions [7,18,40]. However, the main sources of As in the atmosphere are anthropogenic. The Anthropocene period has been proposed to have caused global-scale contamination of the biosphere through atmospheric dispersion of As [40]. Metal smelting (copper, zinc, and lead) and coal combustion are the main anthropogenic sources of As [41]. Coal contains both inorganic and organic forms of As. During coal combustion, ashes are produced and deposited in soils and water [18]. In addition, breathing air with high levels of As can cause lung damage, shortness of breath, chest pain, and coughing [27].

Preliminary studies indicated that combined exposure to atmospheric and groundwater arsenic could significantly increase health risks due to carcinogenic and non-carcinogenic effects [42].

## 4. Anthropogenic As, a Global Environmental Problem with Health Risks

This section has been dedicated to the main contributions of anthropogenic activities to environmental pollution. Anthropogenic activities have contributed and continue to play a significant role in the release of As into the environment. With some exceptions, inorganic As forms are usually more toxic than organic arsenicals, and the trivalent oxidation state is more toxic than the pentavalent oxidation state. In general, at least in humans and many animals, the hierarchy in toxicity of inorganic and organic arsenicals is DMA(III), MMA(III) > As(III) > As(V) > DMA(V), MMA(V) >TMA. The major pentavalent products DMA(V) and TMA (as TMAO) are approximately 100-fold and a 1000-fold, respectively, less toxic than As(III) [10,23,43,44,45]. Besides the chemical species and concentration, As toxicity is also related to bioavailability, and therefore with the rate at which it is metabolized and the degree of bioaccumulation in tissues and cells [44,46]. All of the local physio-chemical, geochemical, and biological factors, as well as anthropogenic activities, determine the great geographical differences of As contamination around the world [4,24,43,47]. Globally, about 200 million people are exposed to potentially toxic levels of As, making this a relevant and extensive public health problem. This metalloid has been classified in the Group 1 of carcinogenic compounds for humans by the International Agency of Research on Cancer (IARC, 2004) [48]. Exhaustive research has demonstrated that both acute and chronic exposure to As caused diverse and severe human disorders, that have been extensively reviewed in recent years [49,50,51,52]. Due to the high toxicity of this element, the World Health Organization [53] established 10 μg/L as the maximum safe level in drinking water in its provisional guideline for As, which is in accordance with the suggested total 15 μg of inorganic As intake per kilogram of body weight. Human exposure to As can take place via ingestion (oral), dermal contact, inhalation, and even parenteral routes [54]. The main causes of As ingestion by humans is to drink water contaminated with this metalloid or to eat certain contaminated foods as fishes or crops (especially As hyperaccumulator plants, as rice) that, in most cases, were irrigated with groundwater containing As [55,56,57]. It is estimated that nearly 108 countries are affected by As contamination in groundwater, with concentrations beyond the recommended maximum permitted amounts by the World Health Organization [55]. The most serious As contamination of aquifers has been found in Brazil, Australia, Afghanistan, India, Bangladesh, Vietnam, and Cambodia [4,43,58]. 

We want to emphasize the role of As as an environmental and food chain contaminant. As we denote above, it is well-documented that human exposure occurs both by drinking water containing As and by consumption of food of both terrestrial and aquatic origin [23]. As bioaccumulation and trophic transfer in both freshwater and the more studied marine ecosystems, are not well understood. Aquatic organisms play important roles in As speciation and cycling in marine and freshwater environments [59]. As is an abundant chemical element in marine waters, and its average concentrations tend to be less variable than those of freshwaters [17]. In marine ecosystems, most studies showed that molluscs and shellfish could accumulate more As, followed by crustaceans and fish, revealing the tendency of no biomagnification of inorganic As in the food web [60,61]. The main hypothesis, from experimental results, is that the inorganic As present in seawater is taken up by phytoplankton and other organisms at lower trophic levels. These primary producers and consumers are preyed on by other marine animals, causing As to be transformed to organoarsenic species and biomagnified through the food chain. Arsenobetaine (AB) is the predominant organoarsenic species found in most finfish and shellfish, typically accounting for more than 90% of the total As [14,57,62]. Fortunately, the most toxic inorganic As species are accumulated in greater quantity at lower trophic levels in the food chain. Recent studies reveal that benthic habits were an important factor for As biomagnification in marine ecosystems [56]. Little is known about As bioaccumulation in organisms and biomagnification in freshwater ecosystems and the results are disperse and controversial [57]. Experimental data from As contaminated lakes showed an enhanced trophic transfer of As through the base of the aquatic food web in weakly stratified lakes. In these lakes, there is greater As bioaccumulation than in stratified lakes with similar levels of contamination [63]. 

Consumption of rice grains from plants cultivated in arsenic-contaminated agroecosystems is the second cause of human As poisoning. Rice is the most important food for more than 50% of the world population. This cereal is mostly cultivated under flooded paddy soil conditions. The As speciation and plant availability in the paddy soil environment is controlled by different biotic and abiotic factors [3]. However, the biogeochemical behavior of As in paddy soil–rice systems makes it easily available for plant uptake and subsequent accumulation in rice grains [64,65]. Several studies in rural areas contaminated with As, located all over the word, indicate that As accumulates in some parts of the rice plants, such as the roots, shoot, rice husk, and in the rice grains [66,67,68], and As accumulation in paddy roots was 28- and 75-fold higher than in shoots and rice grains, respectively [66]. The relative distribution of the organic and inorganic As species among different rice cultivars varies depending on the geographic origin, the rice-growing condition and level of contamination [69]. Some authors, analyzing many samples from Bangladesh, India, and Europe stated that As(III) is the dominant arsenic species in rice grain, followed by As(V) and (DMA) [68,69,70]. In any case, it is clear that As contamination on rice agroecosystems produces many adverse effects in humans [71], animals, plants, and soil microbiota [72]. At present, several mitigation strategies are being developed, applying diverse technological/biotechnological approaches in order to reduce this serious global health risk [64,69].

## 5. Microbial Biotransformations: Impacts on Arsenic and Arsenic Methylation Cycles

Toxic metals and metalloids have exerted selective pressure on life since the rise of the first organisms on earth. As is a ubiquitous element that has a very ancient origin. According to some authors [73], life has been exposed to the toxic metalloid As since the rise of the first organisms, approximately 3.5 Ga, during the Archean. Concentrations of As in marine sedimentary iron formations and shales of this period, suggest early oceans were very rich in As. This geochemically derived inorganic As would have existed primarily as trivalent As(III), that later can be partially transformed in As(V), due to the atmospheric oxygenation [74]. Therefore, first microorganisms have evolved to tolerate/resist moderate or high concentrations of As, and even some of them can obtain energy from the respiration of this metalloid [15]. From this evolutionary point of view, it is not surprising that some microorganisms, mainly prokaryotes, play fundamental roles in the recycling of this element, and genes coding enzymes involved in As transformations became widely distributed in the microbial world [75,76,77]. These biotransformations are focused on producing As resistance or, alternatively, on obtaining energy for growth from this element. As is continuously recycled through the lithosphere, atmosphere, hydrosphere, biosphere, and anthroposphere. Some of its stages and connections are not well understood at present. Microorganisms play an important role in all major transformations involved in As recycling. It is relevant to know the molecular mechanisms involved, as their optimization (i.e., biomethylation, bioaccumulation) can significantly contribute to reducing As pollution in certain environments. The main stages or processes of the As cycle are as follows:

### 5.1. Oxidation and Reduction 

Although there are only two ecological relevant species of inorganic As, As(III) and As(V), the microbial transformations involved in As biorecycling are complex. As we stated above, As(III) is more toxic than As(V) in most of biological systems. Numerous heterotrophic and chemolitoautothrophic microorganisms present the enzymatic activity arsenite oxidase (AioBA), which catalizes the oxidation of As(III) to the less toxic species As(V) [78]. The first arsenite oxidase was purified from *Alcaligenes faecalis* in 1992 [79], and later this activity was detected in *Herminiimonas arsenicoxydans* and in a strain of *Rhizobium* sp. Nowadays, homologous sequences to the gene *aioBA* have been identified in some species include in α-, β-, γ-Proteobacteria, Actinobacteria, Aquificae, Bacteroidetes, Chlorobi, Chloroflexi, Crenarchaeota, Deinococcus-Thermus, Firmicutes, and Nitrospira [29]. This gene is co-transcripted with various genes *ars*, which provide As(III) resistance in these prokaryotes [80]. Two different physiological groups can reduce As(V), with distinct purposes. First, some chemolithoautotrophic bacteria (such as *Geospirillum arsenophilus*, *Alkaliphilus metalliredigenes*, *Sulfurospirillum barnesii*, *Desulfotomaculum auripigmentum*, etc.) are able to use As(V) as a terminal acceptor of electrons [75,76]. This process is denominated dissimilatory reduction of As(V) and it is a singular anaerobic respiration, that contributes to the generation of As(III), and thus, to As mobilization. Arsenate-respiring prokaryotes are a phylogenetically diverse group that can be easily isolated from anaerobic environments, which indicates that they are active in certain anaerobic environments, particularly groundwater and sediments [77]. This physiological capacity is not exclusive of Bacteria, but it is also shown by several Archaea, for instance by *Pyrobaculum arsenaticum*, an obligate anaerobic, hyperthermophilic arsenic-respiring prokaryote, that can also respire selenate [81]. In the other microbial group that can reduce As(V), the main objective is As resistance, doing so by As(V) uptake, intracellular reduction to As(III), and later expulsion to the environment with an efflux pump [76]. The genetic systems involved in this mechanism of As resistance are the *ars* operons. These genetic systems are widely distributed in bacterial and archaeal species [82]. In some prokaryotes, this system provides resistance to inorganic arsenic [As(V)]), but in other species, the *ars* operons have evolved to contain additional genes that confer the capacity of As methylation that additionally increase the arsenic resistance spectrum to organoarsenicals [83,84]. The location of this type of As resistance genes is on plasmids, transposons, and genomic islands and denotes the involvement of horizontal gene transfer processes.

### 5.2. Biomethylation As Methylation Cycle

A relevant part of As biogeochemistry is the processes of As methylation and demethylation, which constitute the As methylation cycle [85]. In aquatic and soil ecosystems, As exists mainly as inorganic As(V) and As(III) and some methylarsenicals, products of microbial methylation (Figure 3).

Moreover, in waters, some organic arsenicals are found in fish and shellfish (arsenobataine, arsenocholine, dimethylarsinic acid (DMA), monomethylarsonic acid (MMA)). We can also find methylated As in mono-, di-, or trimethylarsines, which are less toxic than As(III) [88,89]. The trimethylarsine (TMA(III)) form is almost nontoxic at moderate concentrations and can be volatilized [89]. As biomethylation is a mechanism with a wide distribution in nature, many microorganisms (bacteria, archaea, fungi, protists), plants, animals, and humans present this physiological capacity [88]. At least in photosynthetic microorganisms (microalgae and cyanobacteria), the biological purpose of inorganic As biomethylation is controversial. Some researchers consider biomethylation as a detoxification mechanism while other authors reject this role, since trivalent methylated species (MMA(III) and DMA(III)) are more toxic than the precursor iAs species ([22] and references contained in it). There are two main As methylated compounds generated by microorganisms; methylarsenite (MAs(III)) and arsinothricin (2-amino-4-methylarsonobutanoic acid), an arsenic-containing amino acid with antimicrobial activity, which inclusion in the group of antibiotics has recently been proposed [90]. In this section, we will focus on prokaryotes, the most studied microorganisms with this metabolic ability. In bacteria, As methylation can be carried out by some aerobic and anaerobic bacteria [91]. The main mechanism was detected in more than 120 bacterial species and was further characterized in *Rhodopseudomonas palustris* [92]. This biomethylation process is mediated by the enzyme arsenite S-adenosylmethionine methyltransferase (ArsM, AS3MT in animals), which converts the inorganic trivalent arsenic As(III) into mono-, di-, and trimethylated species [93]. Although certain methylarsenicals are more toxic than As(III), cells do not accumulate these compounds; instead they can detoxify them, using several pathways [94,95]. MAs(III), the first product of this route, can be oxidized to MAs(V) in presence of air. DMA(III) is the likely second product, but its instability in air results in rapid oxidation to DMA(V) under aerobic conditions [95]. MMAs(V) and DMA(V) have often been detected in vegetative tissues and grains of rice plants [69]. However, no *arsM* orthologs have been found in higher plants, only in microalgae, so the main source of the methylated arsenic species appears to be microbial in origin [96]. A second prokaryotic As biomethylation pathway has been elucidated generating TMA(III) in several anaerobic bacteria (for instance, *Clostridium collagenovorans*, *Desulfovibrio vulgaris*, and *Desulfovibrio gigas*), and arsine in the archaea *Methanobacterium formicium*, as end products of As methylation. Experimental data from aerobic and anaerobic prokaryotic species, that commonly inhabit soil environments, have concluded that encoding a functional ArsM enzyme does not guarantee that a microorganism will actively drive As methylation in the presence of the metalloid [97]. Besides the microbial arsenic methylation, humans and other animals, which possess the enzyme AS3MT, can contribute to methylarsenicals generation in the environment. In the generally accepted classical pathway, inorganic trivalent As(III) is a preferential substrate by human As(III) S-adenosylmethionine (SAM). As(III) is reduced to As(V) and by successive oxidative methylation, in which the mono-, di-, and trimethylated pentavalent arsenic species are formed before, the respective trivalent species are generated [88].

### 5.3. Immobilization and Liberation of Arsenicals 

Although we explain in more detail these particular aspects elsewhere in this review, microorganisms also contributed to the arsenical cycling with two additional processes. Many bacterial and phytoplankton species, and even soil microorganisms, use arsenobetaine as compatible solute and to protect cells against extremal temperatures. Thus far, two pathways have been proposed for the biosynthesis of this compound. The first one postulates the formation of arsenobetaine from di- or tri-methylated arsenosugars that are primarily produced by eukaryotic organisms at the bottom of the aquatic food chain. The breakdown of these organoarsenicals lead either to the formation of arsenocholine as an intermediate that then could be further oxidized to arsenobetaine, or to the synthesis of dimethylarsinoyl-ethanol, which could serve through several biotransformation reactions as a precursor for arsenobetaine production. The alternative route for arsenobetaine synthesis proposes dimethylarsenite as the starting compound [11,12,13]. When these cells die they release this As form and other organoarsenicals to water or soil. These amounts might be quantitatively relevant under some circumstances, for instance in microbial blooms.

## 6. Arsenic Toxicity in Eukaryotic Microorganisms: Main Effects and Targets

To understand the mechanisms of toxicity and detoxification of As in eukaryotic microorganisms, it is necessary in the first place to know the As uptake pathways in these microorganisms. Two main As(IIIl) uptake routes have been described in yeasts; by means of aquaglyceroporyns and through hexose transporters [98]. In most biological systems, As(III) uptake is undertaken via the transporter proteins aquaglyceroporins (AQPs). Moreover, these proteins allow the transport of water, non-polar solutes such as urea or glycerol, the reactive oxygen species hydrogen peroxide, and gases such as ammonia, carbon dioxide, and nitric oxide, and other metalloids such as Sb(III) [99]. Aquaglyceroporins have also been shown to be a major route of bidirectional movement of As(III) into and out of cells in eukaryotes (and also in bacteria), including humans [100,101]. The first characterized eukaryotic AQP involved in As(III) entry was Fps1p from the yeast *Saccharomyces cerevisiae.* Fps1p is a plasma membrane glycerol channel with a critical role in osmoregulation. Its main physiological role is the regulation of intracellular level of glycerol in response to changes in osmolarity. Inactivation of Fps1p results in enhanced cellular tolerance to As(III) and Sb(III). On the contrary, cells expressing a hyperactive Fps1p protein are highly As(III) and Sb(III) sensitive [100]. Additionally, under laboratory conditions, there are more than 20 glucose permeases in *S. cerevisiae* that can transport As(III) and MAs(III) into cells. This mechanism is usually less efficient than the uptake mediated by aquaglyceroporins [98,99,100]. Like AQPs, these sugar transporters, which physiologically are responsible for hexose uptake, are bidirectional [101]. As(V) is chemically similar to phosphate (Pi) and enters into most cells by Pi transporters [101]. Inorganic As uptake in microalgae presents similar mechanisms to those in yeasts as it is mediated by protein transporters embedded in the plasma membrane. As(V) crosses plasma membrane through phosphate (Pi) transporters and As(III) makes its ways into algal cells via hexose permeases and (aquaglyceroporins) channels [55,102,103]. In the presence of As, besides the concentration and the environmental factors that influence its speciation (Eh, pH, and so on), some water characteristics (temperature, light intensity, and expose duration) can influence the As uptake and metabolic pathways [56,102]. In particular, phosphorus concentration in water is very important. Algal cells take in As(V) through phosphate transporters due to the similar properties of As(V) and Pi, and numerous experimental data support a competition between them for intracellular transport. Likewise, phosphate concentration also affects the accumulation, biotransformation (i.e., As(III) oxidation), and excretion of As species [104,105].

Data about the adverse effects of inorganic As species on eukaryotic microorganisms are really scarce in comparison with those from mammals. Nevertheless, it is noteworthy that several targets of metal toxicity and tolerance mechanisms in unicellular eukaryotes appear to be quite similar to those in higher eukaryotes, so the microbial studies in these eukaryotic microorganisms, might prove useful for identifying similar mechanisms in higher eukaryotes. In yeasts (such as *S. cerevisiae*), As(III) causes adverse effects at three main cellular levels: (1) reactive oxygen species (ROS) generation, (2) protein misfolding and aggregation, and (3) inhibition of DNA repair [98]. Experimental results showed that sodium arsenite inhibited yeast cell growth, and the inhibitory effect of cell growth was positively correlated with As(III) concentrations. In addition, As(III) caused loss of cell viability in a concentration- and duration-dependent manner in yeast cells [106]. This cellular death by apoptosis has been associated to high levels of intracellular ROS [107]. Many mitochondrial processes are targeted by arsenicals [108,109] and As(III) inhibits ATP synthesis in yeast mitochondria, because of mitochondrial membrane potential decrease in exposed cells [107]. Moreover, As(III) disrupts the actin and tubulin cytoskeleton in yeast, and probably interferes with folding of de novo synthesized actin and tubulin monomers [109]. The CWI (Cell Wall Integrity) pathway is important for protecting yeast cells against cell wall stress induced by pentavalent As through its upregulation of genes involved in cell wall biosynthesis that leads to cell wall architecture remodeling [110]. From the study of 75 sensitive and 39 resistant mutants against As(III), Johnson et al. [111] reported that protein damage is the key mode of action for As(III) toxicity. As sensitive mutants contain altered genes involved in protein translation, signal transduction, regulation of transcription, and iron homeostasis. On the contrary, the matching genes in the resistant mutants are overrepresented by ribosomal genes and genes involved in protein translation [111]. Likewise, it is demonstrated that the ubiquitous toxic metalloid arsenic, such as As(III), inhibits efficiently the rapamycin-sensitive TORC1 (TOR complex 1). It is well-known that the conserved Target Of Rapamycin (TOR) growth control signaling pathway is a major regulator of genes required for protein synthesis [112]. The molecular connection between iron homeostasis and As toxicity (As(V)) was corroborated later by analyzing the *S. cerevisiae* genome-wide response to As(V) by DNA microarrays. The genes of the *Fe* regulon constitute an important component of the As(V) genomic response, and the arsenic also disrupts iron uptake [113]. Nuclear Envelope Budding (NEB) is a recently discovered alternative pathway for nucleocytoplasmic communication. In *S. cerevisiae*, NEB comprises a stress response aiding the transport of protein aggregates across the nuclear envelope cellular stresses. In the budding yeast, the process is induced after heat shock, hydrogen peroxide, As(III) exposure, and proteasome inhibition [114]. In relation with DNA, it has been demonstrated in yeast that As presents a direct genotoxic action, as well as an indirect action by generating oxidative DNA damage and inhibition of DNA repair [98]. The molecular mechanism and adverse cellular effects resulted as consequence of As exposure in microalgae are not well understood. In fact, there is only a few studies focused on these topics.

Comparative analysis of As(V) and As(III) toxicities in microalgae indicated that great differences not only exist in the tolerance levels among the strains/species, but also in which As species (pentavalent or trivalent) is the most toxic inorganic As form. Indeed, the statement [115] that marine microalgae are more sensitive to As(III), while freshwater algae are more sensitive to As(V), is not true since there are many exceptions. Moreover, in some species both inorganic As forms present the same biotoxicity ([116,117] and references within). In microalgae, ROS generation is also associated with As toxicity. Environmentally relevant concentrations of As(V) caused increased ROS level in *Chlamydomonas reinhartii* [118]. In *C. acidophila*, superoxide generation levels presented significant differences depending on the two As inorganic forms. Under As(III), the most toxic form for this strain, there was a directly proportional relationship between the superoxide increment and the As(III) concentration while ROS generation was significantly lower for As(V) treatments [117]. In these photosynthetic microorganisms, the main adverse effects of As were detected in thylakoids, stigma, and mitochondria; i.e., in organelles indirectly and directly involved with energy generation (see Figure 4).

Moreover, lipid and starch energy reserves were also affected [119,120]. For instance, transcriptomic studies showed that *Scenedesmus* sp. remodeled its cellular composition in presence of As(III) and As(V) by accumulating a significant quantity of lipid at the expense of photosynthetic pigments, carbohydrates, and proteins [119,121]. Some evidence supports the biosynthesis of metabolites, such as lipids, and carbohydrate storage can be promoted by abiotic stresses and/or heavy metal/metalloids exposures, possibly through the induction of ROS accumulation [122]. In the ciliate *Tetrahymena thermophila*, unlike many other eukaryotic microorganisms, As(V) is much more toxic than As(III). This differential toxicity has been explained by the distinct quantity of ROS generation by both As species. Furthermore, As(V) caused severe mitochondrial damage and induced mitophagy (Figure 5).

## 7. Biotransformation and Resistance/Tolerance to As in Fungi and Protists

All organisms, from bacteria to man, have developed different system to resist/tolerate environmental concentrations of As, which indicates the long evolution of interactions between this metalloid and the biological systems. Essentially, eukaryotic and prokaryotic microorganisms have developed the same As resistance mechanisms (efflux, reduction, oxidation, bioaccumulation, biosorption, etc.), so we cannot say that each group has a specific set of mechanisms. However, the cellular structures, molecules, and molecular machinery (and therefore the genetic support) involved in As resistance present some relevant differences in eukaryotes and prokaryotes. In addition, certain species of bacteria and archaea, which came into contact with As in ancient geological times, have evolved a unique biological capacity to obtain energy by oxidation of As(III) and by anaerobic respiration of As(V). No eukaryotic cell can use As to obtain energy from any biotransformation of As [75,123]. As(III) efflux is probably the oldest mechanism of As resistance, since this valence is the predominant under anaerobic conditions [124]. *Saccharomyces cerevisiae* utilizes several mechanisms to decrease cytosolic As(III) levels and elude the high toxicity of this inorganic As form. Cells may stimulate As(III) efflux through the plasma membrane transporter Acr3p [125]. Alternatively, they may restrict As(III) influx through the aquaglyceroporin Fps1p [126] or conjugate As(III) to the low-molecular-weight thiol molecule glutathione (GSH) and sequester the resulting As(GS)_3_ complex in vacuoles via the ABC (ATP-binding cassette) transporter Ycf1p (see Figure 6) [98,127]. Moreover, during chronic exposure to As(III), it has been proven that *S. cerevisiae* exports and accumulates the tripeptide glutathione (GSH) outside of cells. Yeast cells with increased extracellular GSH levels accumulate less arsenic and display improved growth when challenged with As(III). Conversely, cells defective in export and extracellular accumulation of GSH are As(III) sensitive. Therefore, in this new detoxification mechanism, GSH is exported to protect yeast cells from As(III) toxicity [128]. Besides acting as a metal chelator, GSH also protects cells from metal-induced oxidative damage due to its role in cellular redox control. We will next describe these particular strategies in more detail. According to some authors [98,127,128], the As(III) efflux system based in the arsenite permease Acr3p is the major detoxification pathway in yeasts. Homologues of Acr3 are particularly widespread in archaea, bacteria, unicellular eukaryotes, fungi, and lower plants, but are absent in flowering plants and animals [129]. In most of bacteria and archaea, the transporter ArsB carries out this function. Although ArsB and Acr3 are both As(III) efflux systems, they have important functional and structural differences [101]. The yeast Acr3p is a plasma membrane transporter that confers resistance to As(III), presumably by permitting As(III) extrusion from the cells. Acr3p acts as a low affinity As(III)/H+ and Sb(III)/H+ antiporter. The *acr3* gene is located on a multicopy plasmid conferring resistance to high concentrations of As(III) in *S. cerevisiae* [130]. Additionally, we must remember that aquaglyceroporins are bidirectional channels [99]. In *S. cerevisiae*, the aquaglyceroporin Fps1p is a bidirectional As(III) channel. Prolonged As(III) exposure triggers overexpression of the *fps1* gene, causing the reduction of accumulated As(III), as result of elevated efflux. Surprisingly, the same aquaglyceroporin Fps1, involved in As(III) efflux, is an essential factor to maintain As(V) tolerance in budding yeast [130]. Besides the Acr3p-based system, *S. cerevisiae* has another independent transport system for the removal of As(III) from the cytosol, which includes the enzyme Ycf1p, a member of the ABC transporter superfamily. It catalyzes the ATP-driven uptake of As(III) into the vacuole, also leading to As(III) resistance [128]. Ycf1p is the prototypical yeast ABC transporter with a broad range of xenobiotic and metals/metalloids substrates. In principle, the role assigned to this enzyme is to confer Cd resistance, since the *ycf1* gene is over-expressed in the presence of this metal. Ycf1p is not localized in the plasma membrane, but on the vacuolar membrane [131]. Ycf1p has been shown to contribute to the detoxification of As(III), Sb(III), and several other metals [127,131]. Ycf1p could be a GSH conjugate transporter; it transports GSH-conjugated substrates across the vacuolar membrane, sequestering them within the vacuolar lumen [128].

Yeasts also present strategies to tolerate/resist As(V). Figure 6 shows the main mechanisms of As uptake and resistance in yeasts. *S. cerevisiae* has an arsenate reductase, named Acr2p, which is able to reduce As(V) to As(III), which will consequently be exported outside by the cell. In this reduction, glutathione and glutarredoxin acts as an electron donor [99,132]. *Acr2* gene deletion sensitizes cells only to As(V) [132]. To conclude, we must remark that reduced glutathione is a relevant molecule in As tolerance and oxidative stress in yeasts and many others organisms. In the yeast As response, it can bind to metals/metalloids and the resulting complex is a substrate for proteins that mediate vacuolar sequestration. Secondly, it is an important antioxidant to neutralize the ROS generated by As exposure. Finally, GSH may bind to reactive sulfhydryl groups on proteins (protein glutathionylation), preventing protein oxidation and metal binding [128]. 

In filamentous fungi, surface bonding and vacuole compartmentalization are the main mechanisms of As resistance [133]. Fungal genome sequencing has revealed that many species show homologous genes to bacterial genes encoding arsenite metryltransferases (*Ars M*). Usually, these genes are located in clusters adjacent to other genes encoding As-resistance proteins [16].

Detoxification of As in microalgae (Figure 7) may be achieved by several mechanisms, such as adsorption on cell surface and intracellular biotransformations, including As(III) oxidation, reduction of As(V) to As(III), complexation with thiol compounds, and sequestration into vacuoles [104].

In microalgae, cell walls contain several types of functional groups such as carboxyl, hydroxyl, carbonyl, sulfhydryl, and so on, which are negatively charged and allow entrapment of metallic/metalloid cations. This mechanism, known as adsorption, is not exclusive of microalgae, since it is present in other microorganisms such as fungi and bacteria. Moreover, certain microalgae produce and secrete some polymers (mainly polysaccharides), constituting the so-called Extracellular Polymeric Substances (EPS). Most of these polymers act as polyanions, mediating the adsorption of metals and metalloids cations, including As [134,135]. Biosorption has many potential applications in metal bioremediation. This process does not require intracellular transport, so it can be carried out by both dead and live cells, although the biosorption yield is higher in the latter. In microalgae, the As adsorption could reach 60% of total amount of this element. As(V) was found to be the major arsenic species in cytosolic fractions of microalgae cells, accounting for up to 99% of the total As [136]. It has been postulated that microalgae are able to oxidize As(III) to As(V). However, molecular evidence only supports the existence of this process in prokaryotes [75,137]. In microalgae, such as *Dunaliella salina*, *Cyanidioschyzon* and *Chlamydomonas reinhardtii* no genomic or genetical evidence confirms this statement; there are sole physiological data, complemented with chemical measurements of As species [138,139,140]. On the contrary, there is some molecular evidence supporting the existence of As(V) reduction process in microalgae. In *C. reinhardtii*, two arsenate reductase genes (*CrACR2*) have been characterized and expressed in *Escherichia coli* to study their functional role [141,142]. The experimental data indicate that certain amounts of As(V) are reduced to As(III) and later expelled from the cell. In this freshwater species, both As(V) and As(III) are taken out of the cell [142]. Numerous studies, using different species and strains of microalgae, denote the importance and crucial influence of environmental phosphate concentration on As uptake, toxicity, and biotransformation (e.g., [138]). For instance, in the halotolerant microalga *Dunaliella salina*, the efficiency of As removal by this microalga varies under different phosphate regimes. Thus, in short-term uptake experiments, As(III) or As(V) absorption was significantly suppressed by increased phosphate supply. Under these conditions, oxidation of As(III) to As(V) was also increased [138]. In a soil isolate of *Chlorella* sp., phosphate significantly influences the biotransformation and bioaccumulation of As. As(V) reduction, and thus As bioaccumulation increased, when the alga was incubated in a phosphate-limiting growth medium [143].

Like fungi and other microorganism and organism (except in plants), As biomethylation is a usual strategy of As biotransformation, although its contribution as a detoxification mechanism is controversial. Phylogenetic analysis showed that bacterial *ArsM* is more closely related to fungal *ArsM*, whereas mammalian AS3MT is grouped with eukaryotic algal *ArsM* [92,93]. A crucial strategy for eukaryotic microbial survival in environments with heavy metal pollution is the biosynthesis of metal-binding peptides, that immobilize the metal cations to prevent their interaction with macromolecules (DNA, proteins, lipids) and microbial structures. Metal-binding peptides include genetically encoded metallothioneins (MTs) and enzymatically produced phytochelatins (PCs) [144,145]. Metallothioneins are ubiquitous in living organisms and play important roles both in the supply of essential metals to the cell and in metal/metalloid immobilization. Moreover, they might play a relevant function as ROS scavengers. Recently, it was proven that MTs can chelate As(III) in human cells [146], although many previous reports suggested this property [147,148,149,150]. *In silico* genomic analysis showed that *MT* genes are well represented in diverse groups of microalgae, but they are scarce in diatoms and dinoflagellates, two important groups of primary production in marine environments. However, the knowledge of these molecules is still very limited [145]. Phytochelatins are enzymatically produced from glutathione. Uptake of both As(III) and As(V) induced the PCs synthesis in several microalgae [117,148,149,150], although the quantitative importance of this process in As detoxification in these cell is still elusive.

In heterotrophic protists, as the ciliate *Tetrahymena thermophila*, the antioxidant systems dependent on both glutathione and thioredoxin, are relevant in the response against As(V) and As(III). In fact, we have detected that in presence of As inorganic species a strong induction of selected genes encoding for enzymes relevant to these cellular systems occurs. Moreover, under these conditions, there is also over-expression of the genes encoding for certain metallothioneins. The relevance of these non-enzymatic proteins was made evident in knock-down and knock-out strains which become much more sensitive to these As species. Ciliate MTs have unique features when compared to other organisms MTs. These proteins are longer and richer in Cys residues, conferring a larger metal binding capacity, compared to classic MTs [151]. Furthermore, whole cell biosensors of *T. thermophila* with two gene constructs of the *MTT1* and *MTT5* metallothionein promoters and each structural gene, linked with the eukaryotic *luciferase* gene as a reporter, emitted bioluminescence in presence of As(V) confirming the relevance of MTs in As resistance [152].

## 8. Outlook of Eukaryotic Microorganism Applications in As Bioremediation

Arsenic contamination is a global problem which presents severe environmental and health consequences. The potential to apply microbial bioremediation strategies to resolve a localized pollution problem with this metalloid is of great interest. Many aspects should be considered to achieve this using much more complex, but also effective, biological approaches. In recent years, a large number of reviews focused on this topic and more generally on heavy metal remediation using microorganisms have been published (e.g., [135,153,154,155,156,157]). After a detailed examination of the literature, including the references cited herein, we can conclude that microbial bioremediation of As pollution has many potential applications at present. However, many knowledge gaps are yet to be filled before we design efficient technologies. First, some aspects of the multiple mechanisms (particularly volatilization and reduction) involved in As tolerance must be elucidated, such as the molecular bases of the processes and their regulation by environmental and internal cellular factors. There are hardly any genetically modified microbial strains, even of prokaryotes, that they can carry out an optimized process of As bioaccumulation or biosorption under a wide range of environmental conditions. In bacteria, two approaches have been applied to obtain As resistant strains from polluted areas: isolation and growth in general media, and a metagenomic approach consisting of in silico resistance gene search from published microbial genomes which is independent of culture methods [158]. In the exhaustive review by Irshad et al. [156], many experimental attempts of As bioremediation, using bacteria, have been compiled. A new and very interesting technology is the application of microbial fuel cells to remove As in polluted soils, such as rice paddies [159]. As bioremediation studies with fungi or microalgae are scant. In this regard, we stress the necessity of new works from a molecular point of view, in order to design more efficient and robust microbial systems. In natural ecosystems, microorganisms form communities that inhabit in microhabitats. There are intense physiological and molecular interactions among their members that must be analyzed and elucidated to optimize bioremediation processes. From this starting point, System Microbiology might be a new an adequate approach to eliminate As contamination in complex ecosystems like rice paddies. Synthetic biology approaches have been applied recently [160] to copper remediation, which illustrates that microbial bioremediation of As pollution is a feasible feat. 

## 9. Conclusions

After a detailed review and analysis of the microbial interactions with As, in particular with eukaryotic microorganisms, we can conclude:Arsenic contamination is a relevant environmental problem, with global distribution.In most ecosystems, biotransformations of As have been carried out mainly by microorganisms, establishing physiological interactions among them.Eukaryotic microorganisms present many different As tolerance/resistance mechanisms, some of them are applicable in bioremediation.New molecular studies, using eukaryotic microorganisms (microalgae, ciliates, filamentous fungi) are necessary, before developing more efficient strategies of bioremediation.Due to the existence of complex microbial interactions in As polluted ecosystems, Systems Microbiology could be an innovative and appropriate approach to reduce the contamination with this metalloid.

## Figures and Tables

**Figure 1 ijerph-18-12226-f001:**
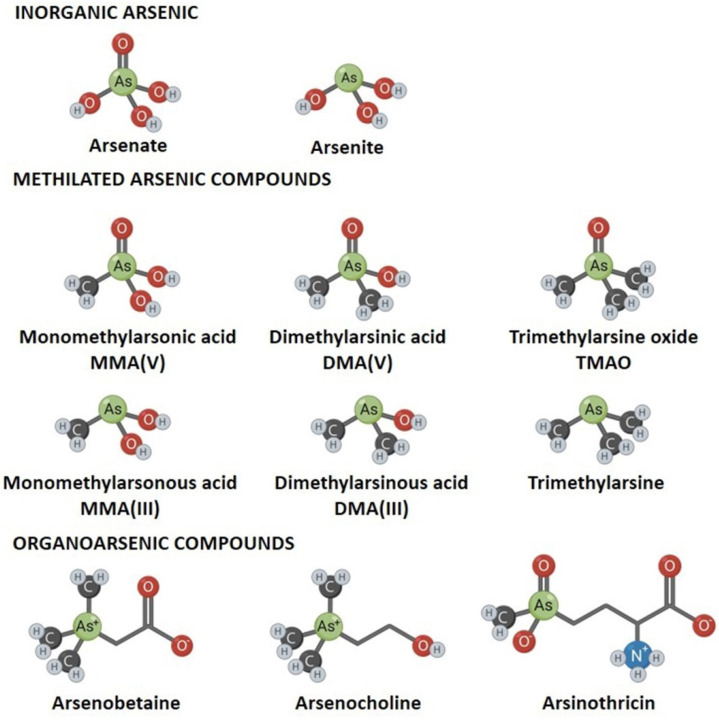
Main chemical forms of inorganic and organic arsenicals.

**Figure 2 ijerph-18-12226-f002:**
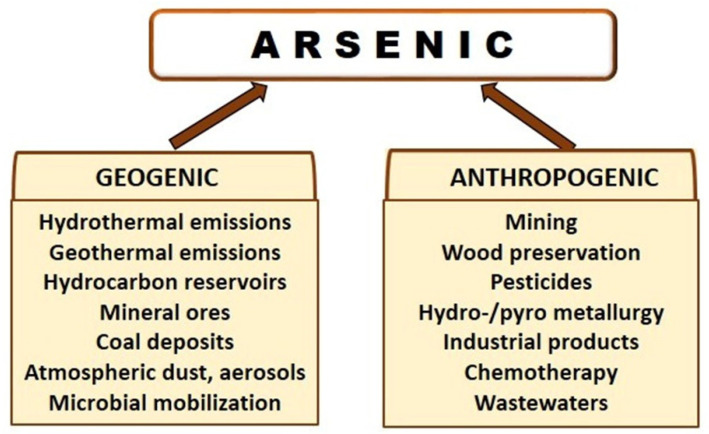
Sources of arsenicals in the biosphere.

**Figure 3 ijerph-18-12226-f003:**
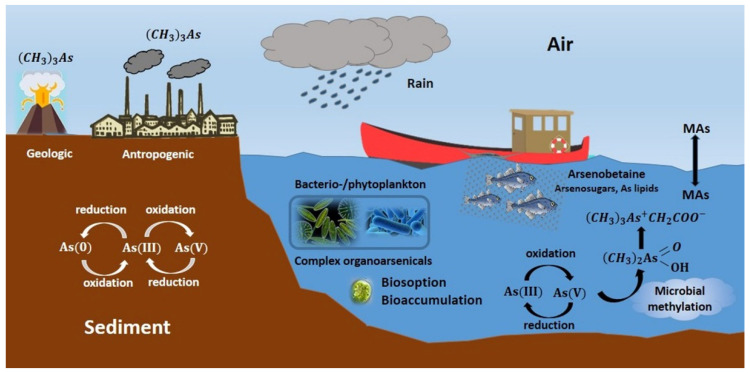
Microbial transformations in the biogeochemical cycle of Arsenic. (CH3)3As: trimethylarsine. Mas: methyl arsenic. Figure inspired from those published by Yüksel et al. and Bhattacharya and Ghosh [86,87].

**Figure 4 ijerph-18-12226-f004:**
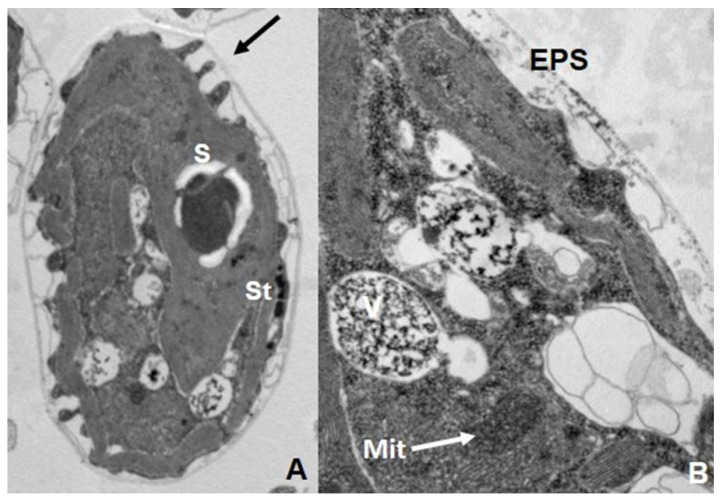
Ultrastructural damage in *Chlamydomonas acidophila* exposed to As(III) 5mM. Note the EPS secretion (arrow), stigma alteration, starch accumulation, and vacuolization. (**A**) General view of a vegetative cell (×20k). (**B**) Detail of a cell, showing mitochondrion degeneration (arrow), vacuoles (×50k). Note the electron-dense content of the vacuoles (V) that corresponds to As (TEM-EDX analysis). Mit: mitochondrion, St: stigma, S: starch, EPS: Extracellular Polymeric Substances.

**Figure 5 ijerph-18-12226-f005:**
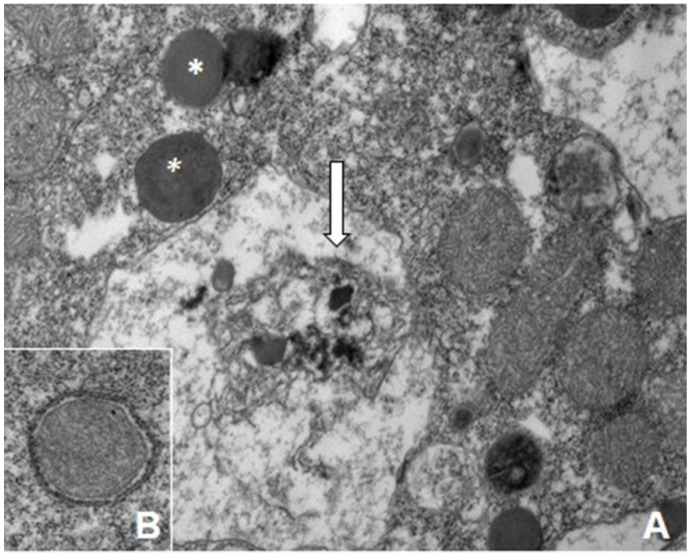
(**A**) Detail of the cytoplasm from a vegetative cell of the ciliate *Tetrahymena thermophila*, exposed to As(V), 30 μM, 24 h, showing mitochondrial degeneration by mitophagy. Degraded mitochondria (*). Arrow points to an advanced autophagosome (×25k). (**B**) Detail of an early mitoauto-phagosome (×25k).

**Figure 6 ijerph-18-12226-f006:**
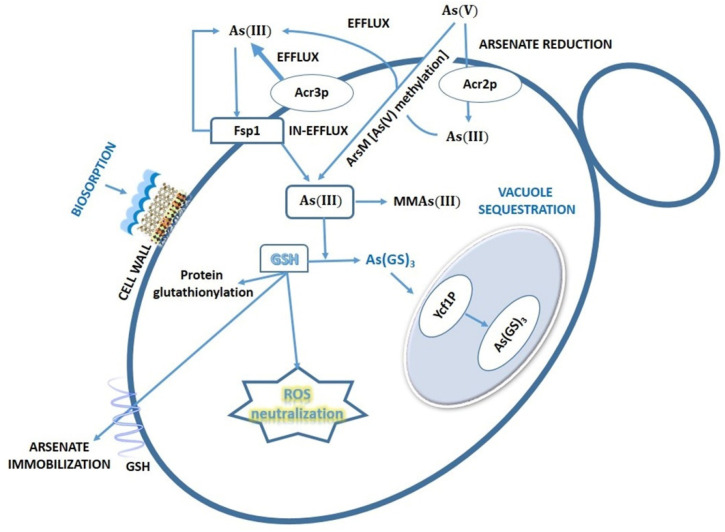
Uptake and resistance mechanisms to As in yeasts. MMAs(III): mononomethylarsonic acid. GSH: glutathione. As(GS)_3_: arsenic triglutathione.

**Figure 7 ijerph-18-12226-f007:**
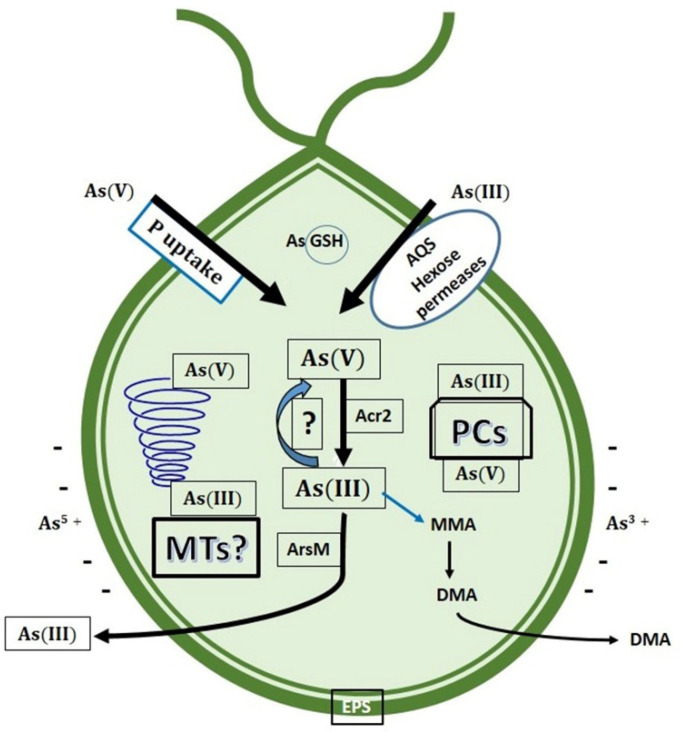
Uptake and resistance mechanisms to As in microalgae. MTs: metallothioneins. PCs: phytochelatins.

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
