# Peer review of "Interactions with Arsenic: Mechanisms of Toxicity and Cellular Resistance in Eukaryotic Microorganisms"

_ijerph, 2021, doi:10.3390/ijerph182212226_

Round 1
Reviewer 1 Report
Dear authors, this manuscript is a valuable review on some issues of the environmental impact caused by arsenic compounds. I consider that it is well written, and the choice of bibliography is appropriate. I find it suitable for the journal, pointing some minor corrections before its possible acceptance:
1) I miss some information on the methods followed to develop the review (databases used as information sources; keywords, dates or other filters used…). Herein I provide a direct link with the PRISMA guidelines for reviews, where guidelines for methods are detailed:
http://prisma-statement.org/PRISMAStatement/Checklist.aspx
2) The review focuses on eukaryotes, although aspects of prokaryotes are sometimes named. For any reader who is not familiar with this topic, I consider that it would be positive for this article to detail at some point in the text the main differences between both types of organisms in relation to the toxicity and resistance to arsenic.
3) I would appreciate if the size of the atoms in molecules of Table 1 were bigger (hydrogens are especially difficult to see).
4) Not all the arsenic compounds mentioned in the text are shown in figure 1, such as methylarsenite or arsinothricin. Thus, could it be possible to provide the structure of all arsenic compounds mentioned in the text, either in figure 1 or in new figure/s? I would appreciate it as a reader who is not familiar with these compounds, but this is just a suggestion.
5) Throughout the text, there is a combined use of the word arsenic and its abbreviation (As), so I suggest to unify the use of the abbreviation, except in those points where this word is to be highlighted.
6) The use of capital letters at the beginning of some words must be revised in the manuscript. For example, in line 29 use E (capital letter) for Earth; or in lines 31-32 use lower case for arsenic and copper.
7) The first sentence of section 3 must be revised.
8) Page 5. I suggest to divide the text in some paragraphs to enhance readability.
9) Lines 220-222. Provide the reference of the study/es that achieved the results mentioned in this sentence.
10) Line 356. The name of this species could be abbreviated.
11) Line 41. Superscripts and subindex of numbers in chemical compounds must be revised (see lines 41, 322…).
12) This is just a reminder. If any of the images has been taken from any source, it must be duly indicated.
Author Response
Dear Reviewer
Thank you very much for your comments and contributions, which have allowed us to correct several errors in the original version. We have also provided new information from the literature to clarify some doubts in some sections. The changes made will be reflected in the attached version 2 of the manuscript (V2).
Response to comment 1
The manuscript is not a meta-analysis, but a personal review and update of some less known aspects of As interactions with selected groups of eukaryotic microorganisms. For that reason we do not include keywords, dates, filter, etc. However, we can specify some of these data below, if you are interested.
Data bases: Scopus, PubMed, Science Direct.
Keywords: We have introduce multiple combinations of the following terms: arsenite, arsenate, arsenic, toxicity, genotoxicity, recycling, yeast, fungi, microalgae, bacteria, bacterial, review, resistance mechanisms, biomagnification, bioamplification, immobilization, metallothioneins, glutathione, phytochelatins, As sorption, biosorption, efflux, influx, arsenite oxidase, arsenate reductase, methylation, rice, plants, paddy fields, bioremediation, and so on.
Date: Until 30 September 2021.
Response to comment 2:
The reason why we have not added a comparative table on resistance mechanisms and toxicity levels as suggested, is because it is very complicated to establish quantitative reference data on toxicity/resistance between large groups (Eukaryotes vs. Prokaryotes), because there are variations even between strains (Ecotypes/Phyllotypes) of the same taxonomic species. Regarding the overview of resistance mechanisms, they are essentially the same in both domains (Eukaryotes and Prokaryotes), although there are more in prokaryotes. Moreover, the same resistance/tolerance mechanism involves different proteins (and therefore different genes and operons) and/or different structures. However, in version 2 of the manuscript, we have added new paragraphs and literature reviews that may clarify this point.
Response to comment 3 and 4:
Thank you for these comments. We have included your suggestions in figure 1 which has been modified. We hope that it has improved considerably.
Response to other comments:
As you will see in version 2 of the manuscript, we have corrected some of the errors you have pointed out, we have divided some sections into paragraphs to make them easier to understand, and we have improved some of the images, citing the sources used.
We hope that you will find version 2 a significant improvement on the original manuscript.

Reviewer 2 Report
The text is very well referenced and detailed. I suggest revising the subtitle names for shorter, more objective expressions.
Author Response
Dear Reviewer
Thank you very much for your comments and contributions which have allowed us to correct several errors in the original version. Also, we have provided new information from the literature to clarify some doubts in some sections. The changes made will be reflected in the attached version 2 of the manuscript (MINOR_V2).

Reviewer 3 Report
The paper entitled “Interactions with Arsenic: Mechanisms of toxicity and cellular resistance in eukaryotic microorganisms” is a deliberate review of the literature concerning biogeochemical cycle of arsenic with special emphasis of threat to public health, mechanisms of microbial detoxification and potential bioremediation of polluted areas. This publication is well written and clearly organized. The information on different ways of arsenic in biochemical cycles are gathered in a synthetic way.
I my opinion the paper is a valuable contribution to the understanding of the role of microorganisms in mitigation of this metalloid in the contaminated environment.
I recommend to accept the publication in a present form after minor correction as follows:
Line 2: “arsenic” instead of “Arsenic”
Line 41: replace “AsH3” with “AsH3”
Line 75: replace “As0” with “As0”
Please correct words in Figure 2: “hydrothermal” instead of “hydrothermal”, “geothermal” instead of “geothermical”, “hydrocarbon” instead of “hidrocarbon”, “mobilization” instead of “movilization”
Line 207: why “Atmospheric” in capitalic?
Line 322: replace “As3+” with “As3+”
Line 344: why “Temperature” in capitalic?
Line 540: “groups of microalgae”
Author Response
Dear Reviewer
Thank you very much for your comments and contributions which have allowed us to correct several errors in the original version. Also, we have provided new information from the literature to clarify some doubts in some sections. The changes made will be reflected in the attached version 2 of the manuscript (MINOR_v2)

Reviewer 4 Report
The review addresses a crucial environmental issue, analysing and proposing detoxification mechanisms. The proposed work could be a valuable approach in recognizing and studying the mechanisms for environmental remediation.
However, for a better understanding of the topics addressed, an introductory paragraph clarifying the structure would be useful.
Some comments on the paragraphs will be given below.
- Arsenic, a metalloid with complex chemistry and biogeochemistry
The first paragraph on the chemical complexity of arsenic and its presence in biogeochemical flows is well structured and provides consistent and relevant information.
- Arsenic sources and releases in Biosphere
The second paragraph is poor in content, the information reported can be merged into the previous paragraph, modifying the title by adding the part on release into the atmosphere.
- Antropogenic As a global environmental problem with health risks
This paragraph is valuable, it is advisable to add, if available, additional information and bioaccumulation data on the rice cultivars mentioned. Would it be possible to table the results even if they derive from bibliographic sources?
- Microbial biotransformations. Impacts on arsenic and arsenic methylation cycles
This section is well organized. However, the aim of the explanation of such cycles is not clear enough.
- Arsenic toxicity in eukaryotic microorganisms. Main effects and targets
The paragraph dealing with the ways of absorption of arsenic is detailed. It explains very well the mechanisms that are started in the presence of arsenic in the cells, in the different species examined. If possible it would be desirable to include color photographs.
- Biotransformation and resistance/tolerance to As in fungi and protists
It is advisable to better explain the influence of environmental phosphate on the ability of cells to absorb As, as specified in line 525.
- Outlook of eukaryotic microorganism applications in As bioremediation
The paragraph seems more a conclusion. It is advisable to combine this paragraph with the conclusions.
Author Response
Dear Reviewer
Thank you very much for your comments and contributions, which have allowed us to correct several errors in the original version. We have also provided new information from the bibliography to clarify some doubts in some sections. The changes made will be reflected in the attached version 2 of the manuscript (V2).
Response to comment 1
We have prepared an introductory text, explaining the content and structure of this review.
Response to comment 2
We have changed the second paragraph, adding a release part to the atmosphere.
Response to comment 3
Sorry, but we have decided do not include additional information of As bioaccumulation in rice cultivars. This technical information was published everywhere else in the literature and besides, it is out of the scope of this microbial review.
Response comments 4-6:
As you will see in version 2 of the manuscript, we have corrected some of the errors you have pointed out, we have divided some sections into paragraphs to make them easier to understand.
We hope that you will find version 2 a significant improvement on the original manuscript.
Response to comment 7
In our opinion, this part of the review is very important for future applications, so we consider that it should not be combined with the conclusions.
In this section we have summarised the recent previous reviews on the topic "As bioremediation" and in addition, we suggest new technological approaches to solve As contamination using microorganisms. This part has to be short because we did not want to compile and describe previous reviews on As remediation/bioremediation, which also includes other metals or metalloids.
